# Coherent Vibrational Anti-Stokes Raman Spectroscopy Assisted by Pulse Shaping

**DOI:** 10.3390/molecules30102243

**Published:** 2025-05-21

**Authors:** Kai Wang, James T. Florence, Xia Hua, Zehua Han, Yujie Shen, Jizhou Wang, Xi Wang, Alexei V. Sokolov

**Affiliations:** 1Department of Physics and Astronomy, Institute for Quantum Science and Engineering, Texas A&M University, College Station, TX 77843, USA; jtflorence6@physics.tamu.edu (J.T.F.); zehuahan@tamu.edu (Z.H.); wjz1993@physics.tamu.edu (J.W.); xwangphy@gmail.com (X.W.); 2Innovation Academy for Precision Measurement Science and Technology, Chinese Academy of Sciences, Wuhan 430071, China; huaxia@apm.ac.cn; 3Laboratory of Neurotechnology and Biophysics, Rockefeller University, New York, NY 10065, USA; yshen01@rockefeller.edu

**Keywords:** Coherent anti-Stokes Raman scattering, pulse shaper, spectral focusing

## Abstract

Coherent anti-Stokes Raman scattering (CARS) is a powerful nonlinear spectroscopic technique widely used in biological imaging, chemical analysis, and combustion and flame diagnostics. The adoption of pulse shapers in CARS has emerged as a useful approach, offering precise control of optical waveforms. By tailoring the phase, amplitude, and polarization of laser pulses, the pulse shaping approach enables selective excitation, spectral resolution improvement, and non-resonant background suppression in CARS. This paper presents a comprehensive review of applying pulse shaping techniques in CARS spectroscopy for biophotonics. There are two different pulse shaping strategies: passive pulse shaping and active pulse shaping. Two passive pulse shaping techniques, hybrid CARS and spectral focusing CARS, are reviewed. Active pulse shaping using a programmable pulse shaper such as spatial light modulator (SLM) is discussed for CARS spectroscopy. Combining active pulse shaping and passive shaping, optimizing CARS with acousto-optic programmable dispersive filters (AOPDFs) is discussed and illustrated with experimental examples conducted in the authors’ laboratory. These results underscore pulse shapers in advancing CARS technology, enabling improved sensitivity, specificity, and broader applications across diverse scientific fields.

## 1. Introduction

Coherent anti-Stokes Raman scattering (CARS), appropriately first reported from the Ford Motor Company by P.D. Maker and R.W. Terhune in 1965, is a third-order optical nonlinear process that has proven to be one of the most powerful nonlinear spectroscopy techniques for probing vibrational and rotational modes of molecules with high specificity and sensitivity [1]. For CARS of molecular vibration, the signals provide a distinct chemical “fingerprint” of the molecular composition. For CARS of molecular rotation, it provides a unique spectral tool to study molecular structure in gas-phase molecules. In the rest of this paper, we will focus on CARS based on molecular vibration, which has advantages in biophotonics and chemical identifications. A typical CARS process involves three beams (pump, probe, Stokes) interacting with a medium to generate a coherent signal at the anti-Stokes frequency $\omega_{AS}=\omega_{pump}+\omega_{probe}-\omega_{stokes}$, or $\omega_{AS}=2\omega_{pump}-\omega_{stokes}$ when the pump is also used as the probe beam (Figure 1). The signal is resonantly enhanced when the frequency difference between the pump and Stokes beams matches a molecular Raman resonance, providing intrinsic vibrational contrast. Compared to spontaneous Raman scattering that suffers from inherently weak signals due to its reliance on inelastic scattering, CARS benefits from coherent interactions that enhance the signal-to-noise ratio (SNR) by 10^4^–10^5^ times, significantly reducing spectrum acquisition time [2,3]. CARS offers several advantages over other Raman techniques. Unlike spontaneous Raman and other coherent Raman (stimulated Raman scattering, and coherent Stokes Raman scattering, or CSRS), the blue-shifted CARS signal minimizes fluorescence interference. Phase matching conditions among the pump, the Stokes, and the probe in CARS can be more easily achieved and thereby make CARS more advantageous in practice. These features make CARS invaluable for a wide range of applications ranging from gas sensing [4,5,6,7,8,9] to combustion diagnostics [10,11,12,13,14], biological imaging [15,16,17,18,19,20,21,22,23,24], material characterization [25,26,27,28,29], environmental monitoring [30,31], and the food industry [32].

Despite these strengths, detecting a CARS signal is often complicated by the presence of non-resonant (NR) four-wave mixing (FWM) background due to multiple off-resonant vibrational modes and the instantaneous electronic response from other molecules, or even from the same molecules. CARS itself is a specialized form of FWM that targets Raman vibrations. Non-resonant FWM frequencies are generated in the same manner as CARS, albeit not specific to Raman vibrations, thus resulting in a larger range of produced frequencies that do not correspond to specific molecular vibrations of the sample. This can cause the NR FWM background to overpower the CARS signal, and fluctuations of the NR background can also wash out the CARS signal, preventing detection. To suppress or avoid the unwanted NR background and improve signal-to-noise ratio (SNR) of CARS, various techniques have been proposed and employed. Some examples include the following: Epi-CARS detects signals in the backward direction where the NR FWM background is also reduced [33]. This technique requires the size of the sample to be smaller than the optical wavelength. Polarization-dependent CARS suppresses the NR background by leveraging Raman depolarization ratio of certain modes [34]. Interferometric schemes have been employed in detection to increase the SNR ratio and spectral resolution [35,36,37]. Heterodyne detection technique provides an alternative way to suppress the NR background [38]. Notably, combined with dual frequency technique, CARS signal can be obtained with high resolution (4 cm^−1^) over a Raman spectrum spanning more than 1200 cm^−1^ recorded within less than 15 μs [39]. These aforementioned techniques rely on intrinsic properties of samples or a complicated laser system like dual frequency comb. They are less effective in the presence of strong multiple scattering in rough samples because scattering randomizes spectral phases and polarization, which hinders the practical application of CARS.

To circumvent these obstacles, techniques involving the manipulation of laser pulse properties, termed pulse shaping, have been adapted in CARS in the last two decades [40]. A pulse shaper is a versatile tool that can provide an ability to actively tailor the time or frequency structure of optical pulses to precisely meet the needs of the quantum system being manipulated [41,42]. This concept can also include methods developed specifically for optimizing CARS signals, including various approaches for broadband CARS generation, such as the high-speed platform achieving full fingerprint and CH-stretch imaging in biological tissues [22], along with other methods, where recent advances demonstrate label-free chemical imaging of cells and tissues across the full vibrational fingerprint and functional group regions [15]. Both simpler passive pulse shaping techniques (such as imposing linear chirp or spectral filtering) and more sophisticated active pulse shaping techniques (using programmable devices) have been exploited to manipulate the pulses in order to suppress the NR FWM background, optimize the spectral resolution, increase the Raman efficiency, and simplify the experimental setup. The remainder of this article is organized as follows: Section 2 gives a theoretical description of CARS using ultrafast laser pulses. Section 3 introduces passive pulse shaping techniques with two specific examples: hybrid CARS technique, which is established stemming from time-resolved CARS, and the spectral focusing CARS, which is performed by linearly chirping the broadband pulses in order to optimize the light-matter interaction, providing a rapid and selective CARS technique suitable for biological imaging. Section 4 covers active pulse shaping techniques. We firstly review the progress of CARS spectroscopy that relies on a spatial light modulator (SLM). Then we discuss adaptive pulse shaping by using another programmable pulse shaper with the assistance of acousto-optic programmable dispersive filters (AOPDFs), including some potential applications in encryption. Section 5 concludes the article and overlooks the prospect of pulse shaping techniques in CARS.

We are delighted to dedicate this review to our hero Professor Jaan Laane, a colleague and a friend who, through the years, has generously provided general wisdom and numerous insights into applications of vibrational spectroscopy as a tool used to study molecular structure and intramolecular dynamics.

## 2. Theory of Ultrafast CARS

As a third-order nonlinear optical effect, using ultrashort pulses has a significant benefit due to their high intensity and their capability for time-resolved measurements. It leads ultrafast time-resolved CARS spectroscopy where a pair of ultrashort (~50 fs) pulses is typically used to excite broadband Raman modes uniformly, and another ultrashort pulse is used to probe the coherence, resulting in the generation of CARS signal.

As a nonlinear phenomenon, the generation of CARS using ultrafast laser pulses can be described through third-order polarization, which can be written as the sum of the background and resonant contributions [43,44]:(1)$$P_{CARS}^{\left(3\right)}\left(\omega \right)=P_{B}^{\left(3\right)}\left(\omega \right)+P_{R}^{\left(3\right)}\left(\omega \right)=\int_{0}^{+\infty }d\Omega [\chi_{\left(B\right)}^{\left(3\right)}+N\chi_{\left(R\right)}^{\left(3\right)}(\Omega )]E_{pr}\left(\omega -\Omega \right)S\left(\Omega \right)$$(2)$$S\left(\Omega \right)=\int_{0}^{+\infty }d\omega^{\prime }E_{pump}\left(\omega^{\prime }\right)E_{Stokes}^{’}\left(\omega^{\prime }-\Omega \right)$$
where E_pr_(ω) is the probe field; S(Ω) is the convolution of the pump field E_pump_(ω) and the Stokes field E_Stokes_(ω); and N is the concentration of the Raman active molecule. In general, although not always, $\chi_{B}^{\left(3\right)}$ corresponds to the NR response and is purely real while the resonant susceptibility $\chi_{R}^{\left(3\right)}$ is complex and, for the case of Lorentzian lineshape, can be written as:(3)$$\chi_{R}^{\left(3\right)}\left(\Omega \right)=\sum_{j}\frac{A_{j}}{\Omega_{j}-\Omega -i\Gamma_{j}},\;$$
where Ω = ω_pump_ − ω_Stokes_; A_j_, j, and $\Gamma_{j}$ are the amplitude, frequency, and spectral half width of the j th vibrational mode, respectively. The total CARS signal is given by:(4)$$\;I_{CARS}\left(\omega \right)\propto \left|P_{CARS}^{\left(3\right)}\left(\omega \right){|}^{2}=\right|P_{B}^{\left(3\right)}{|}^{2}+|P_{R}^{\left(3\right)}\left(\omega \right){|}^{2}+2Re\left[P_{B}^{\left(3\right)}P_{R}^{\left(3\right)*}\left(\omega \right)\right],\;$$
where the first term $|P_{B}^{\left(3\right)}{|}^{2}$ is the aforementioned NR FWM background that limits the sensitivity of the CARS measurements.

In theory, it is possible to eliminate the NR background by delaying the probe pulse, taking advantage of the instantaneous response of the NR FWM and the longer lifetime of the resonant coherent vibrations. This can be realized in experiments by using an ultrafast laser whose pulse duration is shorter than the lifetime of resonant vibrations. Delaying the probe pulse is effective for background suppression, even when using extremely short (< 10fs) pump/Stokes pulses combined with a sub ps probe to cover ultrabroad bandwidths [45]. However, there is a trade-off in spectral resolution for only using ultrashort pulses. Equation (1) implies that the CARS spectrum is the convolution of the probe field and the third-order susceptibility. The linewidth of the CARS signal peak is determined by the probe bandwidth and the bandwidth of the vibrational mode, whichever is larger. In general, a narrowband probe can help to spectrally resolve multiple CARS lines. However, the bandwidth inherent in the ultrafast pulse is usually broader than the linewidth of Raman lines, so that multiple CARS lines close together could not be distinguished in the spectrum using ultrafast pulses, resulting in multimode interference, generally referred to as quantum beats, in time-resolved CARS spectra. This brings in the time-resolved CARS technique, which can not only identify CARS lines, but also explore dynamics of vibrational states [46,47,48,49,50,51]. Time-resolved CARS suppresses the NR background by delaying the probe pulse, but the technique remains vulnerable to fluctuations. Moreover, when multiple Raman modes are excited together, the multimode interference requires the ability to record high-quality spectral profiles over a relatively large probe-delay span that is challenging in the presence of both scattering and fluctuations.

There are many strategies to optimize CARS generation. A technique named FAST CARS (Femtosecond Adaptive Spectroscopic Techniques for CARS) was proposed for maximizing the coherent molecular oscillation with a sequence of femtosecond pulses and minimizing the NR background for rapid identification of bacterial spore [52,53]. To address a significant part of the Raman vibrational bands simultaneously, a broadband pump and Stokes and a narrowband probe are required [15,22,45]. The narrowband probe can be obtained by laser technique [45] or pulse shaping technique. The broadband spectrum can be achieved from supercontinuum (SC) generation [15,22], which can cover the entire Raman active region (400–300 cm^−1^), or advanced laser technique [45]. When the duration of the broadband pulse is shorter than the period of molecular vibration and the bandwidth is larger than Raman shifts, it can lead to the concept of impulsive excitation [37,40,54] in which a single pulse is used as the pump/the Stokes. It significantly simplifies the experimental setup, but the broadband excitation of Raman bands usually results in strong NR background. Therefore, another selective excitation of Raman is proposed with manipulating both the phase and polarization [55,56].

Pulse shaping technique provides a method to tailor the pulse duration, the bandwidth, the spectral phase, and the temporal delay, aiming to achieve close-to-optimal resonant response with reasonable suppression of the NR background, higher spectral resolution, selective and optimized excitation, and so on. The actual optimal values depend on the specific Raman line width, the sensitivity of a setup employed, and the relative strengths of the resonant and NR susceptibilities. For example, the top-hat spectrum gives a sinc-squared temporal profile as [Sin(ωt/2)/(ωt/2)]^2^ for the probe pulse intensity. Putting the pump and Stokes pulses in the node of the sinc function, as shown in Figure 2, would result in effective suppression of the NR background while the resonant signal remains. Nevertheless, in practice, the background suppression is never perfect, and especially at low concentration of the target molecules the residual background often interferes with the CARS signal. According to Equation (3), we can have *I*_CARS_(*ω*) ∝ $|P_{R}^{\left(3\right)}\left(\omega \right){|}^{2}$ ∝ *N*^2^, which means that in the ideal situation, the intensity of CARS would be proportional to the square of the number of molecules. However, due to the NR background, the interference term $2Re\left[P_{B}^{\left(3\right)}P_{R}^{\left(3\right)*}\left(\omega \right)\right]$ that is proportional to N tends to dominate, resulting in a linear relationship between the signal intensity and the number of molecules. By taking advantage of background interference, a proper tailored probe pulse can lead to a novel heterodyne detection strategy for CARS measurement [54,57].

Pulse shaping techniques can be divided into two categories, passive pulse shaping and active pulse shaping, depending on the energy consuming. Both of these two techniques can be applied to manipulate spectral phase and spectral amplitude, which have various applications depending on laser intensity, laser wavelength, experimental condition, sample, and so on. We will discuss both passive pulse shaping and active pulse shaping in the next two sections.

## 3. Passive Pulse Shaping for CARS

Passive pulse shaping is usually based on apparatuses like optical slit based on knife-edge, glass rods, prism pair, and so on. It is simple to set up and can be used without energy consuming. It provides a straightforward way to manipulate laser pulses, which is implemented in CARS spectroscopy such as hybrid CARS and spectral focusing. We discuss these two techniques in detail.

### 3.1. Hybrid CARS

Hybrid CARS is a technique that combines the advantage of time-resolved CARS spectroscopy with the robustness of frequency-resolved CARS [44,53]. Specifically, besides femtosecond pump and Stokes pulses, a proper designed narrowband probe (it is usually a picosecond pulse) ensures a sufficient spectral resolution that can resolve different Raman modes. The probe can be obtained by a passive pulse shaper, which consists of a set of optics to spatially disperse the component of light into separate wavelengths, an optical slit based on knife edge to narrow the spectrum, and another set of optics to recombine the light. A multichannel detector (usually a CCD) is used to simultaneously record the anti-Stokes signal at all optical frequencies within the band of interest. These features allow us to discriminate between the resonant contribution and the NR background, extracting the CARS signal even at zero probe delay. It is worth mentioning that hybrid CARS can also be realized with advanced laser technology and active pulse shaping.

Figure 2 shows an experiment layout of hybrid CARS based on a homemade pulse shaper using knife-edge technique [58]. The experiment utilizes a Ti:Sapphire regenerative amplifier (Legend, Coherent Co., Saxonburg, PA, USA) with two evenly pumped optical parametric amplifiers (OPAs) (Coherent OPerA-VIS/UV and OPerA-SFG/UV) to output two tunable femtosecond pulses as the pump and the Stokes. A small fraction of the amplifier output is sent through a home-made pulse shaper with an adjustable slit to obtain a probe field with a sinc-squared temporal profile. The three beams are focused separately to make them overlapped at the focuses. The time delay of the pump and the probe are adjusted by two delay stages, respectively. The scattered light is collected and focused onto the entrance slit of the spectrometer with a liquid nitrogen-cooled charge-coupled device (CCD) (Spec 10, Princeton Instruments Co., Trenton, NJ, USA).

The bandwidth of probe that affects the spectral resolution is determined by the width of the optical slit in the pulse shaper [49] (Figure 3). The quantum beats effect occurs when the spectral resolution of the CARS setup cannot resolve both Raman peaks. By narrowing the bandwidth of the probe filed using pulse shaper, the beating pattern gradually transforms from quantum beats to two separate streak lines as shown in Figure 3d. Pyridine molecules that have two Raman modes at 992 and 1031 cm^−1^ are measured using CSRS, which is intrinsically equivalent to CARS, except wavelength (Quantum beats of CARS can be found in Ref. [59]). The two Raman modes are excited simultaneously by ultrashort pulses.

When the spectral resolution is high enough to resolve the Raman shift, the NR background is still there veiling the CARS signal. If modifying the probe field to be a sinc-squared temporal waveform, it can help to further suppress the NR background by adjusting the time delay of the probe. When the first node of the sinc-squared temporal profile overlaps the Stokes and pump pulses, multiple Raman lines stand out against a weak background (Figure 4b), in contrast to the solid (red) curve in Figure 4a, where the probe has the same bandwidth and background dominates. The background is still greater than all the signals because suppression is never perfect. In this work [54], a linear relationship between intensity of signal and concentration, which can be attributed to influence of background as discussed previously, is observed. Also, it highlights a sensitivity that can detect 5 mM glucose in liquid.

In general, hybrid CARS based on a passive pulse shaper is a versatile technique in which both collinear and non-collinear configuration for laser beams can be implemented to measure either transmission or reflection [44,60,61]. It expands the applicability of this technique for studying spectroscopy or imaging of both transparent samples and opaque samples. In a forward scheme to measure transmission, hybrid CARS can perform quantitative measurement and reveal concentrations of target molecules in various environments [45,58,62,63]. For example, organic fluid can be measured in a two-color-beam CARS [45], and glucose in aqueous condition down to 5 mM is demonstrated [58], which can be used for diabetes diagnosis. Molecules in gas phase can be well studied using hybrid CARS based on molecular vibration in the context of molecular concentration [62,63], temperature [64,65,66], and molecular dynamics [66]. Gaseous samples usually have weak NR background, which can ensure a quadratic dependence between CARS signal and the number of molecules [62]. It has been demonstrated to measure the temperature up to 2400 K [64] with an accuracy better than the nano-second laser diagnostical technique. Alternatively, the temperature dependence of vibrational state populations in CO_2_ has been probed using fs/ps CARS to perform thermometry from single spectra [66]. Impulsive CARS can also be adapted in hybrid CARS for thermometry [66,67,68] and bioimaging [69,70]. A narrowband probe light can be achieved using second harmonic generation of the fundamental light assisted by a passive pulse shaper [66], volume Bragg grating [67], laser etalon [68], the phase matching of nonlinear crystal [69,70], and so on. The bandwidth of the probe can be narrowed down to 0.37 cm^−1^ with the assistance of volume Bragg grating, which would be sufficient to study rotational Raman scattering [67]. Reflection measurements might not be able to perform quantitative measurements and are mainly applied in measuring opaque samples like bacteria [71] or microspectroscopy for bioimaging [72,73].

It is worth mentioning that, in addition to vibrational Raman spectroscopy, hybrid CARS for rotational Raman can also be implemented to explore temperature [68,74,75,76,77], concentration [68], and pressure [78] of gaseous samples, particularly for the study of flames and combustion. A technique called second harmonic band compression (SHBC) is developed to generate a narrowband probe field by mixing oppositely chirped pulses [75,76]. Hybrid CARS based on SHBC provides a gas-phase thermometry to diagnose flame and combustion primarily based on linewidth of rotational CARS spectra [74,75,76,77]. The gas pressure can be measured as demonstrated in Ref. [78], where an impulsive excitation is used for rotational CARS, and a single laser shot spectrum of N_2_ is recorded to obtain the pressure.

### 3.2. Spectral Focusing CARS

Another passive pulse shaping technique that has been explored recently is called spectral focusing CARS, in which the quadratic spectral phases of the pump and the Stokes pulses are modified to have the same linear chirp. It can be realized with glass rods which would chirp the pulse passing through [79]. It provides a simple and efficient way to manipulate the spectral phase to achieve picosecond pulses. It reduces power loss compared with other technique to control quadratic phase like using grating-based stretchers [79], chirped mirrors [80], and pulse shapers [81].

By optimizing the length of glass rods, the pump and Stokes pulses are prepared with the same linear chirp, creating a constant instantaneous frequency difference between them that can be tuned to match specific molecular vibrations [79,82,83]. The stretched duration increases the interaction time between the pulses, extending probing of the molecular vibrations at the cost of lower peak power while maintaining the same total pulse energy. The chirped pulses have reduced peak power compared to transform-limited pulses, which consequently reduces the intensities of both NR FWM and CARS, but the contrast of the resonant Raman signal to the NR FWM background is increased according to the experimental results [82,83]. The spectral resolution of spectral focusing can be hundreds of times higher than the un-chirped pulse [84,85]. Spectral focusing CARS is particularly valuable when working with sensitive samples, such as living cells, where high peak powers could cause photodamage. Moreover, the instantaneous frequency difference can be quickly tuned by adjusting the time delay between the pump and the Stokes pulses. This allows for a rapid and selective excitation of narrow vibrational resonances, which is suitable for chemical imaging. If dual frequency comb technique is adapted with spectral focusing, it can further accelerate the imaging speed [86]. These features make spectral focusing CARS highly promising for rapid hyperspectral imaging [87,88,89,90,91,92], particularly in biology and medicine [93,94,95]. It can also be used to explore concentration and temperature for gaseous samples based on rotational Raman scattering by optimizing pulses with 4f pulse shapers in which Raman shifts of 100 cm^−1^ or more are attainable and allow for enhanced detection of high-energy (150–300 cm^−1^) rotational Raman transitions at near-transform-limited optimum sensitivity [96].

We conduct an experiment to demonstrate spectral focusing CARS (Figure 5). A commercial Ti:Sapphire femtosecond (Astrella Co., New York, NY, USA) provides 70 fs pulse at 5KHz (central wavelength at 800 nm) with a pulse energy of 4 mJ. About 40% of the amplifier output is used to pump an OPA to generate Stokes pulses centered at 1040 nm. A portion of remaining 800 nm pulses is used as a pump and probe pulse for CARS. Several glass rods (N-SF 11, dispersion parameter D: −537.38 ps/(nm km) at 806 nm, −214.70 ps/(nm km) at 1040 nm) with a total length of 230 mm are inserted in the pump pulse path to chirp them to around 1.65 ps. Another 350 mm glass rod is inserted in the beam path of 1040 nm beam to stretch the pulse to 1.65 ps. The two pulses are then combined collinearly and focused on the sample to generate FWM, which include CARS and NR FWM. The total intensity of FWM varies as a function of relative time delay, and it would be enhanced when it is on resonant with a Raman mode. Hence, Raman shift can be deduced from the measurement of the light intensity of FWM by varying the time delay instead of measuring the spectra.

In the experiment, we record the spectra of FWM as a time delay between two pulses with a spectrometer (HR550, Horiba Co., Kyoto, Japan). In the data processing, the spectral intensity is summarized at each time delay to obtain the temporal intensity profile. We also record the spectra generated from a piece of glass slide, which is 150 μm, as a reference for the NR FWM background. Samples measured in our experiment are low density polyethylene (LDPE) film, plastic film (polyethylene terephthalate (PET)), and acetone in a cuvette. Figure 6a is the relationship between the measured Raman shift and the relative time between the pump pulse and the Stokes pulse. The data are processed considering the standard spontaneous Raman spectra. The linear fitting implies a linear chirp for both the pump and the Stokes pulses. Figure 6b is the measured temporal profile for LDPE (blue), PET (orange), and acetone (red). They are obtained by summarizing the spectral data recorded by spectrometer at different time delays. It shows that the Raman shift can be linearly correlated with time delay, which is attributed to a linear chirp generated by the glass rod. Even though the higher order chirp is unavoidable in practice, its impact on the result in our experiment is minimal. Figure 6c,d are the retrieved spectra of CARS for LDPE and PET according to the fitting result in Figure 6a. Although the spectral resolution of spectral focusing is insufficient to fully resolve the Raman shifts, the linear chirp enabled accurate mapping of vibrational modes. Discrepancies between the retrieved CARS spectrum and spontaneous Raman spectrum primarily result from the varying non-resonant contributions rather than higher order dispersion, as discussed in Ref. [80]. Our results have shown the potential of spectral focusing CARS as a simple method to achieve high speed imaging in applications such as bio-imaging.

## 4. Active Pulse Shaping CARS

Active pulse shaping usually relies on programmable devices like spatial light modulators (SLMs), acousto-optic modulators, AOPDFs and so on. It can manipulate both spectral phase and spectral amplitude simultaneously, which are implemented for optimizing Raman excitation, selective Raman excitation, and reducing acquisition time. Active pulse shapers based on SLMs and AOPDFs are programmable devices controlled by a computer. There are two different control strategies: open loop control and feedback (adaptive) control [42]. In the open loop configuration, the desired output waveform is specified by the user with knowledge of the input pulse including necessary pre-compensation due to unknown distortion. Alternatively, the adaptive control, also known as a close loop control, relies on the ability to program a pulse shaper under computer control. In this strategy, it usually starts with a stochastically prepared pulse shaper, which would be updated iteratively according to a designed optimization algorithm based on the difference between desired and measured experimental output. The adaptive control scheme is less intuitive, but is suitable for optimization of strongly nonlinear processes, or for manipulation of quantum states with insufficient knowledge. We will cover pulse shaping with programmable pulse shapers such as SLMs and AOPDFs, which are adopted in hybrid CARS.

### 4.1. Pulse Shaping with SLM

An SLM is a widely used programmable device for pulse shaping in optics and photonics. The use of SLMs for programmable phase shaping in CARS was fundamentally demonstrated by Dudovich et al. [40], who showed that a computer-controlled SLM in a 4f pulse shaper enabled selective excitation of vibrational levels and effective suppression of the NR background in single-pulse CARS. An SLM is usually based on liquid crystal, whose pixel size is on the order of µm. The spectral resolution of this device can be close to several wavenumbers. Critically, they achieved high spectral resolution, significantly exceeding the pulse bandwidth limit, by precisely modulating the spectral phase using the SLM to exploit quantum interference pathways [40]. An SLM is commonly integrated into a 4f dispersion management system (pulse shaper, shown in Figure 7). In this configuration, the input laser pulse is spectrally dispersed onto the SLM, which applies a user-defined phase pattern onto the spectral components before they are recombined. Common implementations use programmable liquid crystal modulator arrays that allow independent, simultaneous gray-level control of both spectral amplitude and phase [42]. It provides a simple way to achieve the desired laser pulses, which is valuable to help achieve impulsive excitation CARS [40] and optimization of CARS for microscopy [97,98]. A two beam hybrid CARS combined with a spatial light modulator (SLM) to optimize the Raman excitation has been demonstrated to be able to precisely measure the temperature in a high pressure gas cell based on rotational Raman scattering [99]. Programmable hyperspectral CARS microscopy is achieved by using a 2D SLM to tailor the Stokes light to collect spectral information in a more rapid and efficient manner [100]. In a noise autocorrelation spectroscopy for coherent Raman scattering, an SLM can be used to add noise to the probe beam to achieve a spectral resolution without temporal scanning or spectral pulse shaping [101].

Liquid crystal SLMs can also offer phase-only modulation, which has advantages over other pulse shaping techniques because the power loss is reduced. It brings in unique programmability, which allows for dynamic optimization of complex light-matter interactions inherent in CARS. Phase controlling can lead to various techniques like phase cycling to suppressing the NR background [102], selective excitation of Raman mode [103,104,105], spectral phase optimization for high spectral resolution [106], optimizing CARS with frequency resolved optical gating technique [107], and so on. Temporal pulse shaping, such as generating delayed pulse sequences using an SLM, allows for time-gated detection methods that can suppress the instantaneous NR background, improving the sensitivity for detecting specific molecular in microscopy [103]. A dual-mask SLMs configuration which allows for combined phase and polarization control of the supercontinuum is originally proposed by Oron et. al. [55] and helps to achieve a broadband selective excitation of CARS for microscopy [54,55], manipulation of quantum states [108], thermometry [109,110], stand-off Raman spectroscopy up to a 12 m distance [111,112], and so on. Controlled relative phases using an SLM allows for “all-optical processing”, coherently adding signals from multiple Raman lines of target molecules for enhanced detection or achieving destructive interference to cancel signals from specific (potentially background) molecules [113]. SLM-based shaping enables highly sensitive heterodyne detection in a single-beam configuration, where part of the shaped beam acts as a phase-controlled local oscillator, providing significant signal amplification and linear concentration dependence, ideal for detecting low-concentration species [114]. The achievable spectral resolution in spectral focusing is quantitatively linked to the applied chirp, i.e., the quadratic spectral phase. SLMs enable a precise manipulation of spectral phase. Using SLM-based tailored spectral focusing, Brückner et al. investigated this trade-off, showing experimentally and via simulations how the CARS linewidth could be actively narrowed (e.g., from >50 cm^−1^ down towards the probe limit of ~25 cm^−1^ for acetonitrile) by increasing the chirp magnitude. In principle, it can help to correct much higher order spectral phase [89]. The active control allows the resolution to be optimized for specific molecular features and allows for balancing signal strength against the resolution needed, for instance, to resolve CO_2_ Fermi dyads (~6 cm^−1^) [109] versus broader hot bands (~14 cm^−1^) [66].

### 4.2. AOPDF for CARS

AOPDF is another type of programmable spectral filter that can accomplish similar ends but with a different physical mechanism based on a co-propagating acousto-optic interaction [115,116]. It is capable of manipulating the spectral-temporal phase and amplitude shaping through the interaction between light waves and acoustic waves inside a birefringent crystal. It has been widely used with amplified femtosecond laser system for pulse optimization and stabilization [117,118,119,120], pulse characterization [121,122,123], higher order spectral phase manipulation [124], and selective detection in low frequency Raman imaging [125].

In CARS, we attempt to adopt pulse shaping assisted with AOPDF and a passive pulse shaper to selectively excite certain Raman modes and suppress other Raman lines. We demonstrate a selective excitation of Raman mode by making a molecule like cyclohexane perform “scissors” (Raman shift at 1470 cm^−1^), but no “wagging” (1060 cm^−1^) with its CH_2_ groups of atoms. The experiment is based on the Hybrid-CARS technique with an AOPDF device inserting in the beam path to manipulate the pulse as shown in Figure 8. The device for AOPDF used in our experiment is called dazzler (Fastlite Co., Antibes, France), which can control temporal shapes of total duration anywhere from ~40 fs to ~8 ps and work with wavelength ranged from 1100 nm to 1500 nm. A cuvette filled with cyclohexane is put at the focal point. We use the adaptive control strategy to optimize the pump pulse according to the pre-set Stokes pulse to generate the best experiment results. Genetic algorithm (GA), which has been adopted to optimize CARS with SLMs [126,127], is implemented for the adaptive control of the pump pulse through a computer that remotely controls the AOPDF. The parameters of the spectral phase and the amplitude would be genetic representations of GA. Specifically, GA is implemented through LabVIEW (National Instruments Co., Austin, TX, USA) in the experiment. The fitness function of GA is based on the SNR of CARS at 1470 cm^−1^, which is the Raman mode to be excited. To test our strategy, we pre-set the Stokes pulse with a positive chirp, a negative chirp, and a higher order spectral phase. The best result is shown in Figure 9a,b. Both the pump and Stokes pulses are characterized with cross-frequency resolved optical gating (X-FROG) technique with the assistance of probe light. After 87 generations of GA, the SNR of CARS at 1470 cm^−1^ is around 2000, which is 1000 times higher than that at 1060 cm^−1^. All other Raman modes are suppressed greatly. We also attempt to achieve selective excitation by manually controlling the spectral phase without using a dazzler. The results are similar to those by using GA.

In principle, people can also use pulse shapers to excite all the Raman modes with a flat, uniform pulse in the spectral domain. Selective molecular excitation hinders species-specific detection, while excitation uniformity comes at the expense of excitation strength. These tradeoffs can be clearly traced through theoretical analysis and need to be considered when designing a spectroscopic laser system for a particular goal. The selective excitations could be applied to effectively control the vibration of molecules, which might find applications in biology and chemistry.

In addition, since high order spectral phase cannot be measured by a linear method, we can develop a protocol for encryption, which can only be read through a nonlinear process. In this protocol, the information can be encrypted in the spectral phase of an ultrashort pulse. In order to decrypt the information, we need to generate a 2D spectrogram, which can correlate the spectral intensity to the time and frequency domains. The 2D spectrogram is generated between the encoded pulse and a designed read-out pulse through the four-wave mixing (FWM) process as shown in Figure 8a, which is based on the third-order nonlinearity. We experimentally demonstrate this protocol by using a dazzler to encode the information in the higher order spectral phase. We encode four letters “TAMU” in the spectral phase of a pump pulse whose central wavelength is 1257 nm and bandwidth is around 30 nm (Figure 10b). The read-out pulse is centered at 804 nm with a pulse duration around 200 fs. The experimental setup is shown in Figure 10c. The FWM is generated on a piece of glass by mixing two pulses. The FWM is recorded as a function of time delay between the encrypted pulse and the read-out pulse to generated 2D-spectrogram. The result is shown in Figure 10b where it clearly shows the “TAMU” word in the 2D-spectrogram, but if the pulse is only measured in either the frequency or time domain, we cannot obtain the four letters correctly.

In this protocol, the pulse would be chirped if it propagates in the dispersive medium, but with pre-knowledge of the dispersion, people can design a scheme to compensate for the dispersion and still obtain the right information. This implies that the information can only be read out successfully by a designed read-out pulse, which can have a special design spectral phase as well. If an eavesdropping thief intercepted the encoded pulse, it would still be unreadable. For the data capacity, we can encrypt all the information in one pulse, but for security, we can encrypt the information in a series of pulses as shown in Figure 8. The complexity of the pulse-shape protocol is around 10^200^ in theory, which is sufficiently high for encryption. In Figure 11, the first column shows the decrypted four letters using four different pulses. Their encrypted spectra are shown in the middle column. The last column shows the pulse in the time domain predicted by dazzler.

In addition, programmable pulse shapers such as AOPDF offer a way to manipulate and characterize the spectral phase for ultrafast pulses. It can precisely create higher order spectral phases, which would have many applications in various areas. In the next section, we will discuss a simpler and more practical way, called spectral focusing, to generate equal linear chirp for ultrashort pulses, which can help to achieve selective excitation in CARS spectroscopy.

## 5. Conclusions

In this paper, we review multiple strategies to implement pulse shaping techniques in CARS spectroscopy for molecular vibrations to increase the SNR and spectral resolution. Pulse shaping relies on pulse shapers, including passive pulse shapers and active pulse shapers. Particularly, two typical techniques relied on passive pulse shaping to modify the spectral linewidth (hybrid CARS) and spectral phase (spectral focusing CARS) are comprehensively examined. Furthermore, CARS based on active pulse shaping, which used the programmable devices such as SLMs and AOPDFs, is reviewed. Specifically, a scheme combining an AOPDF and another passive pulse shaper for CARS is experimentally demonstrated for selective excitation and a further application in encryption using ultrashort pulses is introduced as well. Pulse shaping techniques, which can manipulate spectral phase and amplitude of laser pulse, have shown its perspective to effectively optimize the CARS signal generation and detection in a complicated nonlinear optical process. It paves the way for implementing CARS spectroscopy in variety of applications.

## Figures and Tables

**Figure 1 molecules-30-02243-f001:**
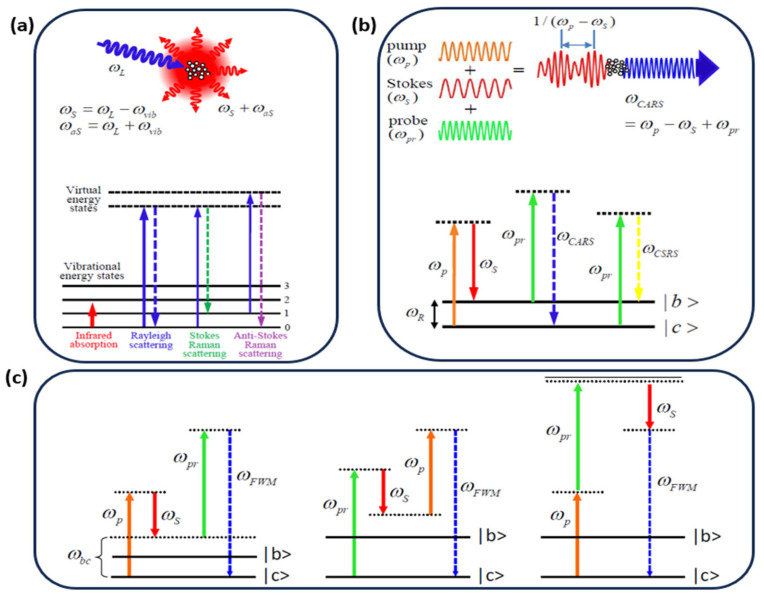
Principle of Raman. (**a**) Spontaneous Raman process. (**b**) Coherent Raman scattering including CARS and coherent Stokes Raman scattering (CSRS). (**c**) Energy diagram for NR FWM background including multiple off-resonant vibrational modes and the instantaneous electronic response. ω_p_, ω_pr_, ω_s_, and ω_as_ are the frequencies of the pump light, the probe light, the Stokes light, and the anti-Stokes light.

**Figure 2 molecules-30-02243-f002:**
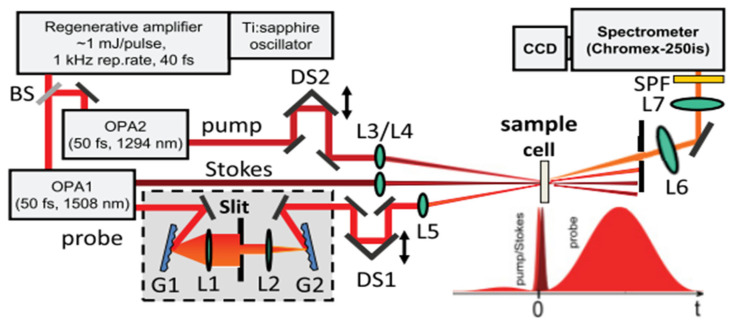
Schematics of the hybrid CARS setup from [58]. BS: beam splitter. OPA: optical parametric amplifier. DS: delay stage. G: grating. L: lens. SPF: short pass filter.

**Figure 3 molecules-30-02243-f003:**
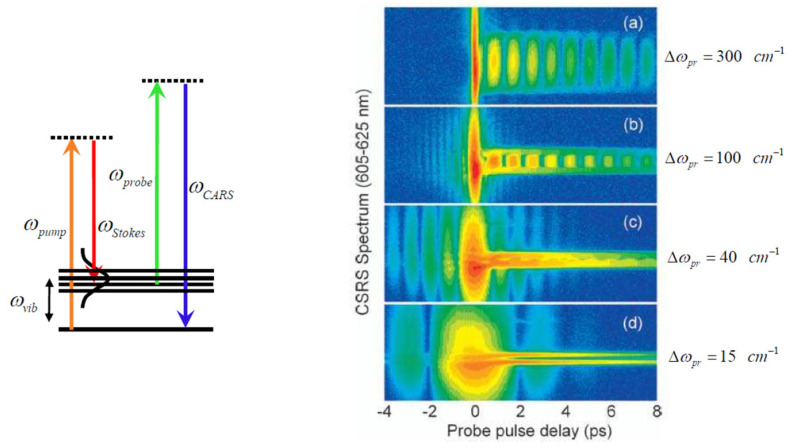
Transition from time-resolved to hybrid CSRS [44]. CSRS spectrograms for different spectral bandwidths of the probe pulse: (**a**) 300 cm^−1^, (**b**) 100 cm^−1^, (**c**) 40 cm^−1^, (**d**) 15 cm^−1^. Two Raman lines of pyridine, 992 and 1031 cm^−1^, are excited via a pair of ultrashort laser pulses. Pump: λp = 737 nm, FWHM ≈ 260 cm^−1^, 0.5 μJ/pulse. Stokes: λS = 801 nm, FWHM ≈ 480 cm^−1^, 0.9 μJ/pulse. Probe: λpr = 577.9 nm, 0.15 μJ/pulse.

**Figure 4 molecules-30-02243-f004:**
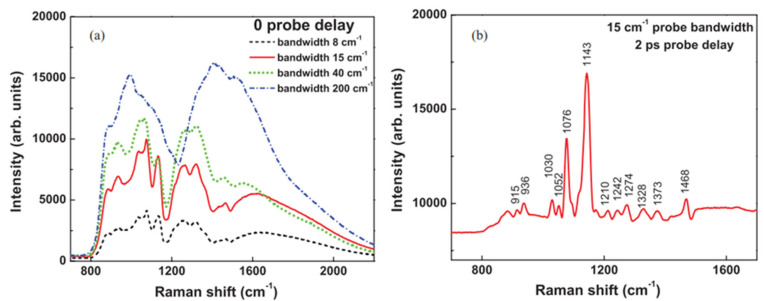
CARS spectra of D-glucose solution at 2680 mM. (**a**) Spectra at 0 probe delay with different probe bandwidths and the same power; (**b**) background suppression when probe is delayed to the node; note that the background level is still around 9000 units. Here the integration time is 0.2 s (From Ref. [58]). Considering the linewidth of Raman modes, the bandwidth of the probe pulse determines if the Raman peaks can be distinguished in CARS. In (**a**), the linewidth of glucose mode is around 20 cm^−1^. Therefore, it can be spectrally resolved by the narrower bandwidth probe, but it cannot be resolved if the bandwidth of probe is broader than 20 cm^−1^.

**Figure 5 molecules-30-02243-f005:**
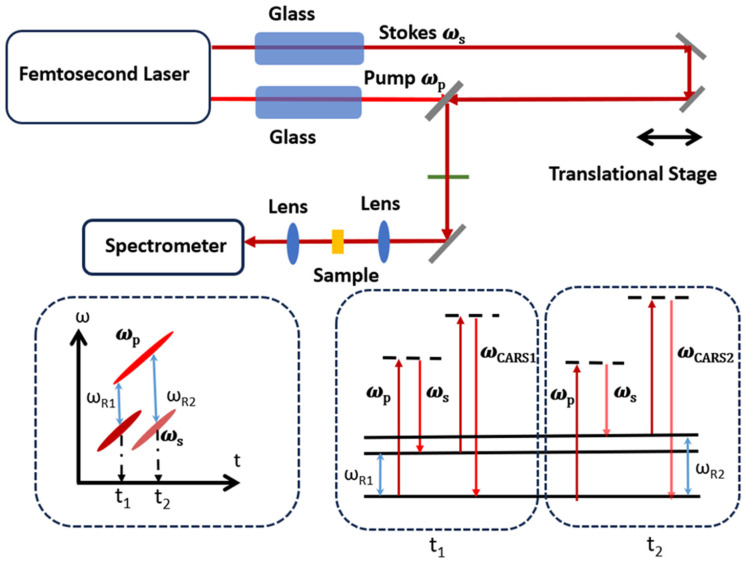
Experimental layout for spectral focusing. Insets are the principle of spectral focusing CARS. The pump field will also be the probe field. At different time delay t_1_ and t_2_, different Raman modes (ω_R1_ and ω_R2_) are excited and different CARS are generated.

**Figure 6 molecules-30-02243-f006:**
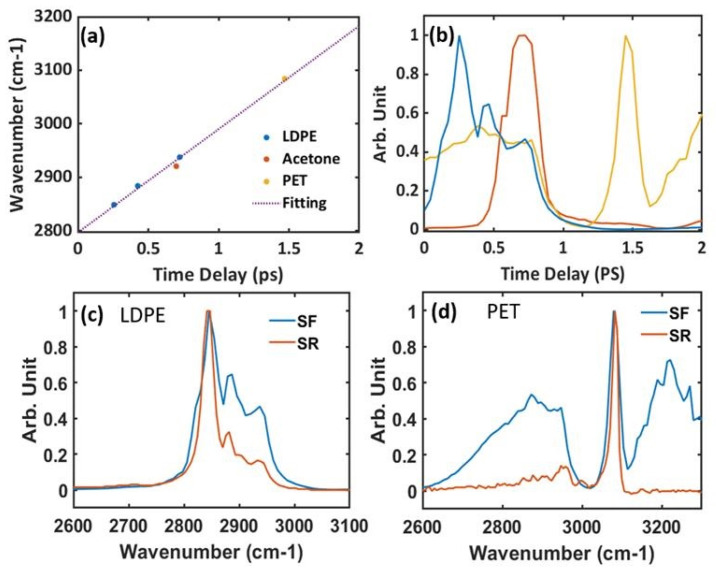
(**a**) The relation between relative time delay and Raman shifts of LDPE (blue), PET (orange) and acetone (red). The Raman shift for LDPE is at 2848.5 cm^−1^, 2884.5 cm^−1^, and 2937.8 cm^−1^. The Raman shift for PET is at 3083.7 cm^−1^. The Raman shift for acetone is at 2921 cm^−1^. The time delay for x-axis is arbitrary and only for reference. It shows a linear dependence between the Raman shift and the temporal delay, which results from the linear chirped pulses. (**b**) Temporal intensity profile for different measurements: LDPE (blue), PET (orange), and acetone (red). It is constructed by recording the data at different time delays. (**c**) Spectrum of LDPE from spontaneous Raman (SR) (red line) and spectral focusing (SF) CARS (blue line). (**d**) Spectrum of PET from spontaneous Raman (SR) (red line) and spectral focusing (SF) CARS (blue line).

**Figure 7 molecules-30-02243-f007:**
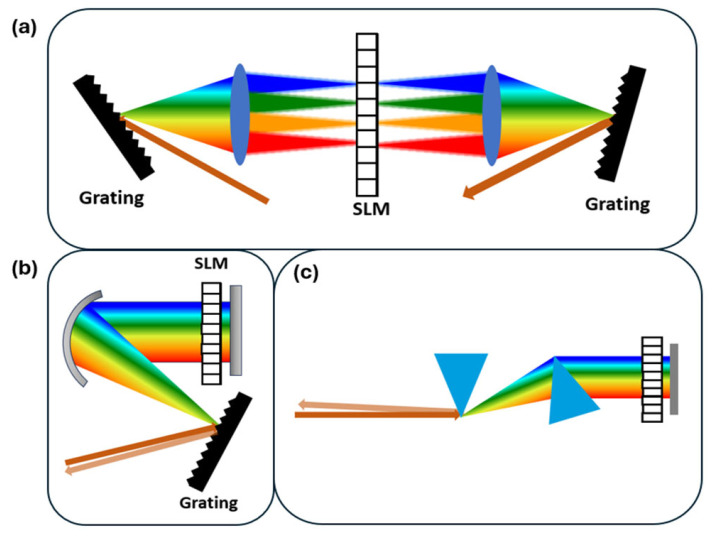
Schematics of pulse shaper with SLM. (**a**) A standard 4f dispersion management system. An input pulse is dispersed by the first grating, collimated by the first lens onto the SLM in the Fourier plane to independently control the spectral phase and amplitude of ultrashort pulses, and then recombined by the second lens and grating after spectral modulation. (**b**) A folding scheme for 4f system. A concave mirror is used to collect the light, and another flat mirror would reflect the beam back. (**c**) Pulse shaper with prism pairs.

**Figure 8 molecules-30-02243-f008:**
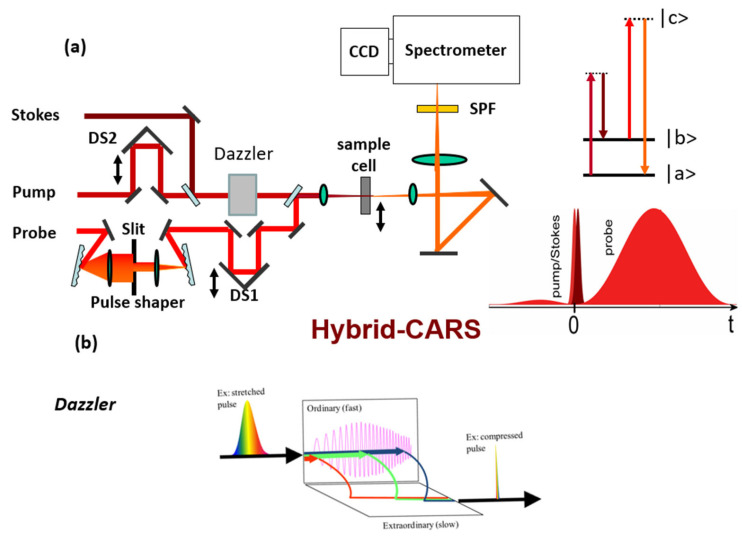
(**a**) Experimental setup for hybrid-CARS with AOPDF. A dazzler is inserted in the beam path to manipulate the Stokes pulse. (**b**) Principle of AOPDF.

**Figure 9 molecules-30-02243-f009:**
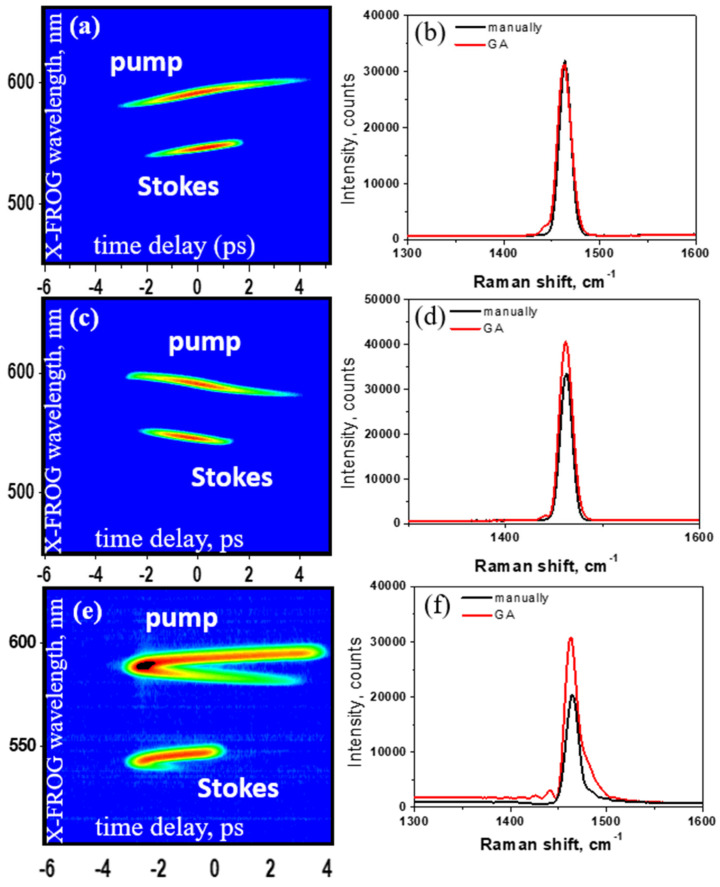
(**a**,**b**) X-FROG of the pump pulse and the Stokes pulse with a positive chirp and the spectrum of CARS. (**c**,**d**) X-FROG of the pump pulse and the Stokes pulse with a negative chirp and the spectrum of CARS. (**e**,**f**) X-FROG of the pump pulse and the Stokes pulse with a higher order spectral phase and the spectrum of CARS. In (**b**,**d**,**f**), black lines are the results using manually control and red lines are the results using GA.

**Figure 10 molecules-30-02243-f010:**
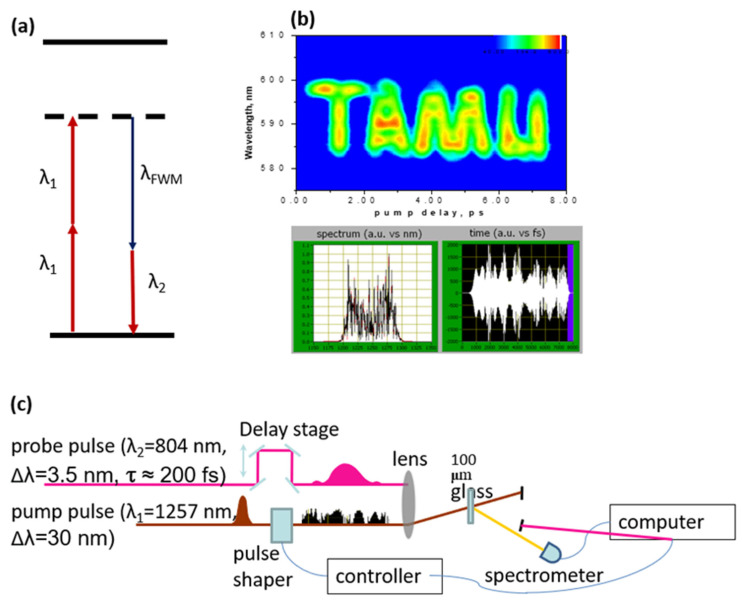
Protocol for encryption based on nonlinear spectroscopy using AOPDF. (**a**) The energy level scheme for FWM. (**b**) The decrypted letters and the corresponded pump pulse in both frequency domain and time domain designed by dazzler. (**c**) Experimental setup.

**Figure 11 molecules-30-02243-f011:**
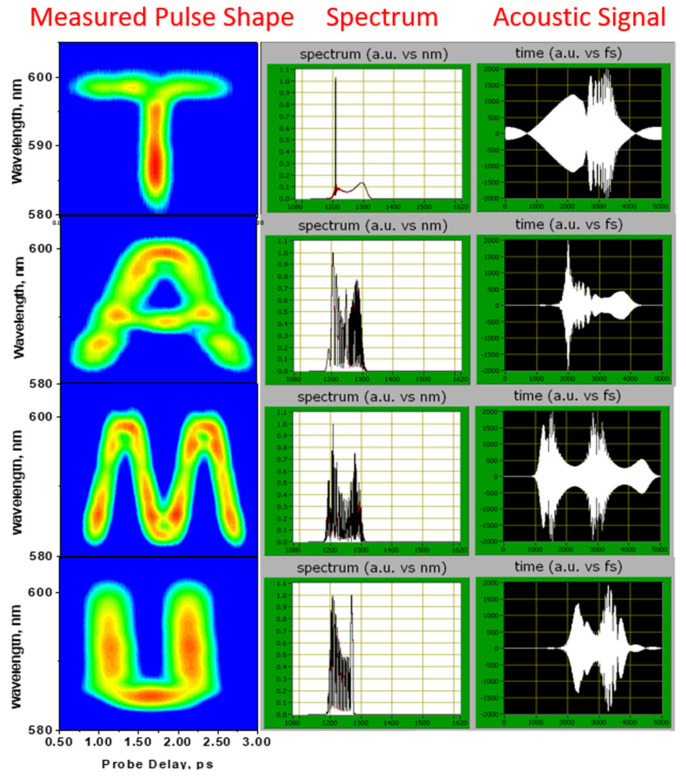
The first column shows the decoded four letters using four different pulses. Their encoded spectra are shown in the middle column. The last column shows the pulse in the time domain predicted by dazzler.

## Data Availability

The data that support the findings of this study are available from the corresponding authors upon reasonable request.

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
