# Peer review of "Coherent Vibrational Anti-Stokes Raman Spectroscopy Assisted by Pulse Shaping"

_molecules, 2025, doi:10.3390/molecules30102243_

Round 1
Reviewer 1 Report
Comments and Suggestions for Authors
This manuscript is a review of the use of pulse-shaping in CARS spectroscopy. I do not find any new insights in this manuscript. I do not see that this manuscript reviews a large or complex field where a nicely organized review paper can save researchers substantial time in understanding the current state of the science and historical development. The authors cover a relatively small number of topics, summarizing a few research directions in CARS where some form of pulse shaping is either nominally or intrinsically used.
I recommend rejection for now. If the author's substantially increase the depth of discussions then I would reconsider but this cursory treatment is not sufficient.
I do not see any mention of the second-harmonic generation compression approach for generation of narrowband pulses from chirped broadband pulse frequency mixing. (i.e., mixing of oppositely chirped arms to create a probe pulse). I do not see any depth in describing optimized pulse shapes for exciting different Raman modes (i.e., in rotational CARS, the concept of impulsive excitation at recurrence periods for amplifying the response of a particular molecule). While the design of the pulse-shape is obvious for pure-rotational transitions, it can become increasingly complex in mixed vibrational ensembles, and there is a whole field dedicated to understanding how to optimize detection through pulse shaping in this scenario. Instead, the authors effectively focus on 'spectral focusing' type selection, but this is effectively just a bandwidth narrowing approach, whereas far more complex approaches can be developed for selective excitation with overlapping ensembles (not just modes that are easily 100 cm-1 apart and fully isolated - where simple instantaneous bandwidth suppression is effective at exciting only one mode).
So at present I find the "review" of this manuscript very superficial at present and cannnot recommend for publication in its present form.
Page 6 middle: Liquid-nitrogen cooled CCD.... could you provide the model number? They exist, but for such an experiment I don't imagine why one would use a liquid nitrogen cooled CCD. Most are water-cooled. or Air-cooled.
Page 3: "this technique requires the sample to be smaller than the optical wavelength"... elaborate here. Unclear what you are saying.
Page 2: "CSRS, whose phase matching conditions might not be satisfied" Elaborate here. In many cases the BOXCRS phase-matching arrangement can easily yield CSRS signals. Not to mention that collinear CSRS is of course phase-matched.
Page 7, "Shutov A. et al. measured the number of O2 moleucles in the gas phase"... very strange citation. Many papers showed O2 detection in hybrid CARS prior to this paper. Like by a decade. And in other CARS arrangements much earlier.
Comments on the Quality of English Language
There are several locations where the English is strange. Page 6 top, "Fig. 2 shows an experimental outlet of hybrid..." outlet? Strange use of English.
Author Response
Reviewer 1
This manuscript is a review of the use of pulse-shaping in CARS spectroscopy. I do not find any new insights in this manuscript. I do not see that this manuscript reviews a large or complex field where a nicely organized review paper can save researchers substantial time in understanding the current state of the science and historical development. The authors cover a relatively small number of topics, summarizing a few research directions in CARS where some form of pulse shaping is either nominally or intrinsically used.
I do not see any mention of the second-harmonic generation compression approach for generation of narrowband pulses from chirped broadband pulse frequency mixing. (i.e., mixing of oppositely chirped arms to create a probe pulse). I do not see any depth in describing optimized pulse shapes for exciting different Raman modes (i.e., in rotational CARS, the concept of impulsive excitation at recurrence periods for amplifying the response of a particular molecule). While the design of the pulse-shape is obvious for pure-rotational transitions, it can become increasingly complex in mixed vibrational ensembles, and there is a whole field dedicated to understanding how to optimize detection through pulse shaping in this scenario. Instead, the authors effectively focus on 'spectral focusing' type selection, but this is effectively just a bandwidth narrowing approach, whereas far more complex approaches can be developed for selective excitation with overlapping ensembles (not just modes that are easily 100 cm-1 apart and fully isolated - where simple instantaneous bandwidth suppression is effective at exciting only one mode).
We sincerely appreciate the referee for reviewing our manuscript and pointing out these concerns. We would like to clarify that our manuscript focuses on pulse shaping techniques used in coherent anti-Stokes Raman scattering (CARS) spectroscopy for molecular vibrations. CARS for molecular rotations is beyond the scope of the current draft. To clarify this, we change the title as “Coherent Vibrational Anti-Stokes Raman Spectroscopy Assisted by Pulse Shaping.” In addition, we emphasize this in the abstract, introduction, and conclusion. Even though we don’t intend to review rotational Raman spectroscopy, we do add a paragraph to remind the readers about this topic. Specifically,we include studies about implementing pulse shaping in CARS for molecular rotations, such as second harmonic band compression (SHBC), optimization pulses for impulsive excitation of CARS, selective excitation of CARS. We add references optimization of rotational Raman using optimized linear chirp pulse, as is a spectral focusing CARS. The new is paragraph:
What is worth mentioning, in addition to vibraitonal Raman spectroscoy, hybrid CARS for rotational Raman can also be implemented to explore temperature [68,74-77], concentration [68], and pressure [78] of gaseous sample, particularly for the study of flames and combustion. A technique called second harmonic band compression (SHBC) is developed to generate a narrowband probe field by mixing of oppositely chirped pulses [75,76]. Hybrid CARS based on SHBC provides a gas-phase thermometry to diagnose flame and combustion primarily based on linewidth of rotational CARS spectra [74-77].The gas pressure can be measured as demonstrated in Ref. [78], where an impulsive excitation is used for rotational CARS and a single laser shot spectrum of N2 is recorded to get the pressure.
So at present I find the "review" of this manuscript very superficial at present and cannnot recommend for publication in its present form.
I recommend rejection for now. If the author's substantially increase the depth of discussions then I would reconsider but this cursory treatment is not sufficient.
We have revised our draft to increase the depth of discussions. As our response to the other referee, we rearrange the draft aiming to illustrating two pulse shaping strategies, passive pulse shaping and active pulse shaping. We combine hybrid CARS (spectral amplitude shaping) and spectral focusing CARS (spectral phase shaping) as the third section for passive pulse shaping. Both of the two parts have been shortened. The section 4 is about active pulse shaping, which is mainly based on programmable pulse shaper such as spatial light modulator and acousto-optic programmable dispersive filters (AOPDF). We add a subsection discussing pulse shaping based on spatial light modulator (SLM). Abstract, introduction and conclusion are revised as well. We add another 33 references to support out manuscript. Please check the manuscript highlighting the modification for more details.
Page 6 middle: Liquid-nitrogen cooled CCD.... could you provide the model number? They exist, but for such an experiment I don't imagine why one would use a liquid nitrogen cooled CCD. Most are water-cooled. or Air-cooled.
The CCD is Spec-10, Princeton Instruments. We add this information in the draft.
Page 3: "this technique requires the sample to be smaller than the optical wavelength"... elaborate here. Unclear what you are saying.
Here it means that the lateral length of the sample is smaller than the wavelength. To clarify this, we modify the sentence as: This technique requires the size of the sample to be smaller than the optical wavelength.
Page 2: "CSRS, whose phase matching conditions might not be satisfied" Elaborate here. In many cases the BOXCRS phase-matching arrangement can easily yield CSRS signals. Not to mention that collinear CSRS is of course phase-matched.
We appreciate this comment and we have deleted this sentence.
Page 7, "Shutov A. et al. measured the number of O2 moleucles in the gas phase"... very strange citation. Many papers showed O2 detection in hybrid CARS prior to this paper. Like by a decade. And in other CARS arrangements much earlier.
We are grateful for referee. We agree that O2 detection by CARS has been studied in decades. However, most of the studies are mainly based on molecular rotations, which are beyond the scope of this draft. The work done by Shutov A. is about using hybrid CARS of molecular vibrations to study molecule concentration. It also clearly shows a nonlinear relation between the concentration and the signal strength due to weak NR background, which are different from that with strong NR background. To make it clearer, we modify the sentences as following: Gaseous samples usually have weak NR background, which can ensure a quadriac dependence between CARS signal and the number of molecules [58].

Reviewer 2 Report
Comments and Suggestions for Authors
The manuscript is a review on pulse shaping techniques for coherent anti-Stokes Raman scattering. The paper has a pleasant flow, concise language, is well structured and merits publication in “Molecules”.
The reviewer’s main concern is about the paper’s motivation and the overall framework. With the title ”Coherent Anti‐Stokes Raman Spectroscopy Assisted by Pulse Shaping”, the reviewer assumes that most readers would expect a thorough review of active pulse shaping techniques, and indeed, the authors define pulse shaping (l86-87) as: “A pulse shaper is a versatile tool that can provide an ability to actively tailor the time or frequency structure of optical pulses to precisely meet the needs of the quantum system being manipulated”. But then, section 2 is dedicated to hybrid CARS using a passive slit, and again section 4 discusses spectral focusing using two glass rods. The presented basic, exemplary experiments obviously have an illustrative purpose, but at the expense of an in-depth discussion of elaborate techniques available in literature. E.g, the group around Markus Motzkus has been highly active in implementing SLMs for CARS spectroscopy (e. g. [A] as one example), and Ref. 86, discussing the influence of various phase functions on the CARS signal, disappears in the reference list. The reviewer also misses Dudovic’s Nature paper, which is probably the most fundamental work regarding pulse shaping for CARS [B].
Two specific remarks:
- The discussion of the achievable resolution could be more quantitative, and application related. (e. g. l134-l155). While Roy et. [C] could resolve the fundamental Fermi dyad in CO2 with active pulse shaping, an appropriate slit size allows resolving further CO2 hot bands for thermometry [D]. The latter reference again points out, that, if pulse shaping includes passive techniques, the number of eligible literature studies increases significantly.
- Pulse shaping in a broader sense should also include the different approaches for broadband CARS, such as Camp et al. [21], which again disappears in the reference list in the introduction, or Vernuccio et al [E]. Even a compressed excitation pulse and a slit-shaped probe allows investigating simple liquids [F], like the experiment for acetone in Fig. 10 in the present paper.
The reviewer therefore highly recommends making the paper’s purpose more evident. As additional experiments are quite unusual in a review paper, the reviewer suggests omitting the two passive setups in section 2 and 4 and instead extending the literature review.
[A] Time-resolved two color single-beam CARS employing supercontinuum and femtosecond pulse shaping, B. von Vacano and M. Motzkus, Optics Communications 2006 Vol. 264 Issue 2 Pages 488-493, DOI: 10.1016/j.optcom.2006.02.065
[B] Single-pulse coherently controlled nonlinear Raman spectroscopy and microscopy, N. Dudovich, D. Oron and Y. Silberberg, Nature 2002 Vol. 418 Issue 6897 Pages 512-4, Accession Number: 12152073 DOI: 10.1038/nature00933
[C] Single-beam coherent anti-Stokes Raman scattering (CARS) spectroscopy of gas-phase CO2 via phase and polarization shaping of a broadband continuum, S. Roy, P. J. Wrzesinski, D. Pestov, M. Dantus and J. R. Gord, Journal of Raman Spectroscopy 2010 Vol. 41 Issue 10 Pages 1194-1199, DOI: 10.1002/jrs.2587
[D] Two-beam femtosecond coherent anti-Stokes Raman scattering for thermometry on CO2 , M. Kerstan, I. Makos, S. Nolte, A. Tünnermann and R. Ackermann, Applied Physics Letters 2017 Vol. 110 Issue 2 Pages 021116, DOI: 10.1063/1.4974030
[E] Full-Spectrum CARS Microscopy of Cells and Tissues with Ultrashort White-Light Continuum Pulses, F. Vernuccio, R. Vanna, C. Ceconello, A. Bresci, F. Manetti, S. Sorrentino, et al. J Phys Chem B 2023 Vol. 127 Issue 21 Pages 4733-4745, Accession Number: 37195090 PMCID: PMC10240501 DOI: 10.1021/acs.jpcb.3c01443
[F] Ultrabroadband two-beam coherent anti-Stokes Raman scattering and spontaneous Raman spectroscopy of organic fluids: A comparative study, T. Koch, R. Ackermann, A. Stoecker, T. Meyer-Zedler, T. Gabler, T. Lippoldt, et al., J Biophotonics 2024 Pages e202300505, Accession Number: 38982549 DOI: 10.1002/jbio.202300505
Author Response
Reviewer 2
The manuscript is a review on pulse shaping techniques for coherent anti-Stokes Raman scattering. The paper has a pleasant flow, concise language, is well structured and merits publication in “Molecules”.
Thank you for your valuable comments on our work. We greatly appreciate your feedback and have made the necessary revisions and improvements to our manuscript based on your suggestions.
The reviewer’s main concern is about the paper’s motivation and the overall framework. With the title ” Coherent Anti‐Stokes Raman Spectroscopy Assisted by Pulse Shaping”, the reviewer assumes that most readers would expect a thorough review of active pulse shaping techniques, and indeed, the authors define pulse shaping (l86-87) as: “A pulse shaper is a versatile tool that can provide an ability to actively tailor the time or frequency structure of optical pulses to precisely meet the needs of the quantum system being manipulated”. But then, section 2 is dedicated to hybrid CARS using a passive slit, and again section 4 discusses spectral focusing using two glass rods. The presented basic, exemplary experiments obviously have an illustrative purpose, but at the expense of an in-depth discussion of elaborate techniques available in literature. E.g, the group around Markus Motzkus has been highly active in implementing SLMs for CARS spectroscopy (e. g. [A] as one example), and Ref. 86, discussing the influence of various phase functions on the CARS signal, disappears in the reference list. The reviewer also misses Dudovic’s Nature paper, which is probably the most fundamental work regarding pulse shaping for CARS [B].
We sincerely appreciate the referee for the concerns. We have modified our title as:” Coherent Vibrational Anti-Stokes Raman Spectroscopy Assisted by Pulse Shaping” to focus on molecular vibrations. We rearrange the draft aiming to illustrating two pulse shaping strategies, passive pulse shaping and active pulse shaping. In detail, we combine hybrid CARS (spectral amplitude shaping) and spectral focusing CARS (spectral phase shaping) as the third section for passive pulse shaping. Both of the two parts have been shortened. The section 4 is about active pulse shaping, which is mainly based on programmable pulse shaper spatial light modulator and acousto-optic programmable dispersive filters (AOPDF). We add the reference suggested by the referee in the draft as:
105 Time-resolved two color single-beam CARS employing supercontinuum and femtosecond pulse shaping, B. von Vacano and M. Motzkus, Optics Communications 2006 Vol. 264 Issue 2 Pages 488-493, DOI: 10.1016/j.optcom.2006.02.065
40 Single-pulse coherently controlled nonlinear Raman spectroscopy and microscopy, N. Dudovich, D. Oron and Y. Silberberg, Nature 2002 Vol. 418 Issue 6897 Pages 512-4, Accession Number: 12152073 DOI: 10.1038/nature00933
110 Single-beam coherent anti-Stokes Raman scattering (CARS) spectroscopy of gas-phase CO2 via phase and polarization shaping of a broadband continuum, S. Roy, P. J. Wrzesinski, D. Pestov, M. Dantus and J. R. Gord, Journal of Raman Spectroscopy 2010 Vol. 41 Issue 10 Pages 1194-1199, DOI: 10.1002/jrs.2587
66 Two-beam femtosecond coherent anti-Stokes Raman scattering for thermometry on CO2 , M. Kerstan, I. Makos, S. Nolte, A. Tünnermann and R. Ackermann, Applied Physics Letters 2017 Vol. 110 Issue 2 Pages 021116, DOI: 10.1063/1.4974030
15 Full-Spectrum CARS Microscopy of Cells and Tissues with Ultrashort White-Light Continuum Pulses, F. Vernuccio, R. Vanna, C. Ceconello, A. Bresci, F. Manetti, S. Sorrentino, et al. J Phys Chem B 2023 Vol. 127 Issue 21 Pages 4733-4745, Accession Number: 37195090 PMCID: PMC10240501 DOI: 10.1021/acs.jpcb.3c01443
45 Ultrabroadband two-beam coherent anti-Stokes Raman scattering and spontaneous Raman spectroscopy of organic fluids: A comparative study, T. Koch, R. Ackermann, A. Stoecker, T. Meyer-Zedler, T. Gabler, T. Lippoldt, et al., J Biophotonics 2024 Pages e202300505, Accession Number: 38982549 DOI: 10.1002/jbio.202300505
We add a subsection regarding pulse shaping based on spatial light modulator (SLM), which would cover content about phase and amplitude shaping as follows:
SLM is a widely used programmable device for pulse shaping in optics and photonics. The use of SLMs for programmable phase shaping in CARS was fundamentally demonstrated by Dudovich et al. [40], who showed that a computer controlled SLM in a 4f pulse shaper enabled selective excitation of vibrational levels and effective suppression of the NR background in single-pulse CARS. SLM is usually based on liquid crystal, whose pixel size is on the order of ?m. The spectral resolution of this device can be close to several wavenumber. Critically, they achieved high spectral resolution, significantly exceeding the pulse bandwidth limit, by precisely modulating the spectral phase using the SLM to exploit quantum interference pathways [40]. A SLM is commonly integrated into a 4f dispersion management system (pulse shaper, shown in Fig. 7). In this configuration, the input laser pulse is spectrally dispersed onto the SLM, which applies a user defined phase pattern onto the spectral components before they are recombined. Common implementations use programmable liquid crystal modulator arrays that allow independent, simultaneous gray-level control of both spectral amplitude and phase [42]. It provides a simple way to achieve the desired laser pulses, which is valuable to help achieve impulsive excitation CARS [40] and optimization of CARS for microscopy [98,99]. A two beam hybrid CARS combined with spatial light modulator (SLM) to optimize the Raman excitation has been demonstrated to be able to precisely measure the temperature in a high pressure gas cell based on rotational Raman scattering [100]. Programmable hyperspectral CARS microscopy is achieved by using a 2D SLM to tailor the Stokes light to collect spectral information in a more rapid and efficient manner [101]. In a noise autocorrelation spectroscopy for coherent Raman scattering, SLM can be used to add noise to probe beam to achieve a spectral resolution without temporal scanning or spectral pulse shaping [102].
Liquid crystal SLMs can also offer phase-only modulation, which has advantages over other pulse shaping technique because the power loss is reduced. It brings in unique programmability, which allows for dynamic optimization of complex light-matter interactions inherent in CARS. Phase controlling can lead to various techniques like phase cycling to suppressing the NR background [103], selective excitation of Raman mode [104-106], spectral phase optimization for high spectral resolution [107], optimizing CARS with frequency resolved optical gating technique [108], and so on. Temporal pulse shaping, such as generating delayed pulse sequences using an SLM, allows for time-gated detection methods that can suppress the instantaneous NR background, improving the sensitivity for detecting specific molecular in microscopy [104]. A dual-mask SLMs configuration allows for combined phase and polarization control of the supercontinuum is origianlly proposed by Oron et. al. [55], which help to achieve a broadband selective excitation of CARS for microscopy [54,55], manipulation of quantum states [109], thermometry [110,111], stand-off Raman spectroscopy up to a 12 m distance [112, 113], and so on. Controlled relative phases using an SLM allows for "all-optical processing”, coherently adding signals from multiple Raman lines of target molecules for enhanced detection or achieving destructive interference to cancel signals from specific (potentially background) molecules [114]. SLM-based shaping enables highly sensitive heterodyne detection in a single-beam configuration, where part of the shaped beam acts as a phase-controlled local oscillator, providing significant signal amplification and linear concentration dependence, ideal for detecting low-concentration species [115]. The achievable spectral resolution in spectral focusing is quantitatively linked to the applied chirp, i.e., the quadratic spectral phase. SLM enable a precise manipulation of spectral phase. Using SLM-based tailored spectral focusing, Brückner et al. investigated this trade-off, showing experimentally and via simulations how the CARS linewidth could be actively narrowed (e.g., from >50 cm⁻¹ down towards the probe limit of ~25 cm⁻¹ for acetonitrile) by increasing the chirp magnitude [89]. In principle, it can help to correct much higher order spectral phase. The active control allows the resolution to be optimized for specific molecular features, and allows for balancing signal strength against the resolution needed, for instance, to resolve CO2 Fermi dyads (~6 cm-1)[110] versus broader hot bands (~14 cm-1) [66].
Two specific remarks:
- The discussion of the achievable resolution could be more quantitative, and application related. (e. g. l134-l155). While Roy et. [C] could resolve the fundamental Fermi dyad in CO2 with active pulse shaping, an appropriate slit size allows resolving further CO2 hot bands for thermometry [D]. The latter reference again points out, that, if pulse shaping includes passive techniques, the number of eligible literature studies increases significantly.
Thank you for bringing this to our attention. We add these information in the manuscript as:
The active control allows the resolution to be optimized for specific molecular features [89], and allows for balancing signal strength against the resolution needed, for instance, to resolve CO2 Fermi dyads (~6 cm-1)[110] versus broader hot bands (~14 cm-1) [66].
- Pulse shaping in a broader sense should also include the different approaches for broadband CARS, such as Camp et al. [21], which again disappears in the reference list in the introduction, or Vernuccio et al [E]. Even a compressed excitation pulse and a slit-shaped probe allows investigating simple liquids [F], like the experiment for acetone in Fig. 10 in the present paper.
We are grateful for this comment. We add discussion of broadband CARS including impulse excitation and selective excitation based on broadband CARS in section 2. The manuscript is revised as:
There are many strategies to optimize the CARS generation. A technique named FAST CARS (Femtosecond Adaptive Spectroscopic Techniques for CARS) was proposed by maximizing the coherent molecular oscillation with a sequence of femtosecond pulses and minimizing the NR background for rapid identification of bacterial spore [52,53]. To address a significant part of the Raman vibrational bands simultaneously, a broadband pump and Stokes and a narrowband probe are required [15,22,45]. The narrowband probe can be obtained by laser technique [45] or pulse shaping technique. The broadband spectrum can be achieved from supercontinuum (SC) generation [15,22], which can cover the entire Raman active region (400-3800 cm-1 ), or advanced laser technique [45]. When the duration of the broadband pulse is shorter than the period of molecular vibration and the bandwidth is larger than Raman shifts, it can lead to the concept of impulsive excitation [37,40,54] in which a single pulse is used as the pump/the Stokes. It significantly simplifies the experimental setup, but the broadband excitation of Raman bands usually results in strong NR background. Therefore, another selective excitation of Raman is proposed with manipulating both the phase and polarization [55,56].
The reviewer therefore highly recommends making the paper’s purpose more evident. As additional experiments are quite unusual in a review paper, the reviewer suggests omitting the two passive setups in section 2 and 4 and instead extending the literature review.
We thank referee for this advice. We have revised the manuscript and divided pulse shaping techniques into passive and active pulse shaping. We think it is better to keep experimental setups for hybrid CARS and spectral focusing. Also, we add a subsection about SLM and one more figure about 4f pulse to show the basic concept. We also extend the literature review by adding a subsection about SLM. We also add a paragraph to briefly discuss rotational CARS based on pulse shaping technique. Please check the manuscript highlighting the modification for more details.

Round 2
Reviewer 1 Report
Comments and Suggestions for Authors
The authors did perform significant revisions to this review article. I appreciate the responsiveness to my direct comments. The review could be of some pedagogical worth to the community. It does not do a comprehensive job of reviewing the uses of pulse shaping to modify vibrational CRS signals, particularly in congested spectral domains, but it does review a couple of use-cases for pulse shaping in vibrational CARS. There is nothing technically wrong/questionable with the manuscript statements any more. I would still implore the authors to make it more useful to the community and review a wider swath of pulse-shaping literature. I will recommend to accept the manuscript and log it myself as a teaching tool for new researchers to pulse-shaping and CARS.
Reviewer 2 Report
Comments and Suggestions for Authors
The reviewer is very grateful to the authors for clearly distinguishing between active and passive pulse shaping now and broadening the scope of this article. In the reviewer’s opinion, this improved the manuscript considerably.
Some typos have been added, please check: e. g. lines 214, 252, 275, 290 (2x), 300.